# The Effects of Green Exercise on Physical and Mental Wellbeing: A Systematic Review

**DOI:** 10.3390/ijerph16081352

**Published:** 2019-04-15

**Authors:** Ian Lahart, Patricia Darcy, Christopher Gidlow, Giovanna Calogiuri

**Affiliations:** 1Faculty of Education Health & Wellbeing, University of Wolverhampton, Walsall WS1 3BD, UK; I.Lahart@wlv.ac.uk; 2Centre for Health and Development, Staffordshire University, Stoke-on-Trent ST4 2DF, UK; patricia.darcy@research.staffs.ac.uk (P.D.); C.Gidlow@staffs.ac.uk (C.G.); 3Faculty of Social and Health Sciences, Inland Norway University of Applied Sciences, Postboks 400 2418 Elverum, Norway

**Keywords:** green exercise, nature, mental wellbeing, physical wellbeing

## Abstract

We aimed to examine the evidence for the proposed additive effect of exercise in the presence of nature (green exercise) by systematically reviewing studies that investigated the effects of outdoor or virtual green exercise compared with indoor exercise. Our review updates an earlier review, whose searches were conducted in April 2010. Trials were eligible if: (a) participants in an outdoor or virtual exercise condition were exposed to views of nature (green exercise); (b) green exercise was compared with indoor exercise with no exposure to nature; (c) included an outcome related to physical or mental health; (d) used comparative or crossover trial design. We searched the following databases from 1st January 2010 to 28th June 2018: PubMed, CENTRAL, EMBASE, PsycINFO, GreenFile, and Sports DISCUS. We assessed risk of bias using the Cochrane “risk of bias” tool. Where possible we conducted a meta-analysis using the inverse variance random-effects method, and where this approach was not possible we presented the results qualitatively and in harvest plots. We identified 28 eligible trials. In a meta-analysis of just three longitudinal trials, the only statistical finding was slightly lower post-intervention perceived exertion with green versus indoor exercise (mean difference: −1.02; 95% confidence intervals: −1.88, −0.16). Compared with indoor exercise, acute bouts of outdoor green exercise may favorably influence affective valence and enjoyment, but not emotion, perceived exertion, exercise intensity, and biological markers. No other consistent statistical differences were observed, apart from a higher enjoyment of outdoor green versus virtual green exercise. We found a high risk of bias across trials and an overall low quality of evidence. In conclusion, there was limited evidence to support the view that green exercise offers superior benefits to exercise without exposure to nature. The low quality of evidence prohibits clear interpretation of trial findings. Future robust and rigorously designed trials are needed to evaluate the effects of long-term and multiple-bout exposure to nature during exercise compared with exercise indoors.

## 1. Introduction

Physical activity is known to provide a wide range of health benefits that can protect individuals from diseases and enhance their mental and physical health [1]. Regular physical activity can prevent and manage a range of chronic conditions, including cardiovascular disease, type 2 diabetes, and certain cancers, and improve musculoskeletal health, weight management, motor skill development in children, and mental health problems [1,2]. However, global estimates show that around one-quarter of adults aged 18 years and over are insufficiently active worldwide [1]. As a result, physical inactivity is one of the leading risk factors for global premature mortality, responsible for 9% of early deaths worldwide [3]. Recommendations for health-enhancing physical activity targeted to different population sub-groups, often make explicit reference to the activity mode, duration, intensity, and frequency. In the past decades, however, the environment in which physical activity take places has emerged as an additional element that can determine the activity’s health benefit. In particular, it has been postulated that physical activity in the presence of nature, a practice also known as green exercise, can provide additional health benefits and, thus, have greater value for preventing disease and enhancing population health in the population [4].

There are several ways in which the added health benefits of green exercise might arise. Simply having better access to natural environments, such as parks, playing fields, or woodlands, provides the space and facilities for physical activity, which may in turn foster a more active lifestyle. Though intuitive, evidence that having good access to natural environments (green and blue space) can promote physical activity is equivocal [5,6,7]. A review of 50 epidemiological studies of objectively measured access to greenspace and physical activity found positive associations in 20 studies, whereas 28 studies offered little support and two reported negative associations [6]. However, this field of research is dominated by studies with cross-sectional design that often prevent the identification of causal relationships between availability of natural environments and increased physical activity in the local population [8,9]. Therefore, a question remains about the possibility of a ‘self-selection’ phenomenon: do natural environments elicit increased physical activity and well-being, or do physically active individuals choose to live in areas with more opportunities for physical activity?

Secondly, there is evidence that people tend to engage in physical activity when in green space and might be active for longer and or at higher intensities in natural environments [10]. For example, the activities that might be well supported by outdoor environments, such as running, hiking, mountain biking, or horse riding, are those that might be undertaken for longer periods of time (compared with indoor activities) [11]. Other studies in trained athletes have indicated that they might be able to exercise at higher intensities in natural environments as they are more distracted from internal signs of fatigue [12] or have lower perceived effort [13,14]. These two effects can interact, resulting in people being more active than they would be in other settings, thus gaining greater health benefits [15].

Finally, being physically active in natural environments may confer additional health benefits, compared with those that would result from the equivalent activity in an urban/built or indoor environment [16,17,18]. This is associated with the notion that exposure to scenes of nature can elicit positive psychological states such as increased positive affect and reduction of psychophysiological stress [5,6,7]. The underlying mechanism linking nature exposure to such psychological outcomes is not yet clear. Possible explanations have included evolutionary perspectives [19,20], elicited feelings of connectedness with nature [21,22], and visual recognition of characteristic features such as the color green [23], and geometrical fractals [24]. Irrespective of the underlying processes, a 2010 review of 25 studies comparing responses to activities (mostly walking or running) in natural versus non-green outdoor built environments or indoor environments found that the former were associated with greater energy and reduced anxiety, anger, fatigue, and sadness [16]. However, by conflating non-green outdoor built environments and indoor environments, as well as exercise and non-exercise conditions, this review might have not taken into account possible confounders such as factors eliciting negative emotional responses (e.g., street traffic) or the acute psychophysiological responses to physical exercise. Thompson Coon et al. [18] reviewed the effects of physical activity in natural environments compared with physical activity indoors on mental and physical wellbeing, health-related quality of life, and long-term adherence to physical activity in 11 studies. The authors reported beneficial effects of natural environments for a range of psychological outcomes, such as revitalization, positive engagement, tension, confusion, anger, depression, and energy. There was also evidence of greater enjoyment and satisfaction with outdoor activity, with indications of greater intent to repeat the activity. However, the review was limited by the small number of papers included, poor methodological quality of the available evidence, and the heterogeneity of outcome measures employed. This made interpretation and extrapolation of the findings difficult.

Given the recent proliferation of work in this area, we have updated Thompson Coon et al.’s [18] systematic review. In line with this previous review, and to explore the potential causal relationship between green exercise and different health outcomes, we restricted our search to studies with experimental or quasi-experimental designs. Specifically, we sought to address the following research questions: (1) Are the longitudinal effects of exercising in an outdoor natural environment (green exercise) different to exercising indoors without exposure to nature (indoor exercise)? (2) Are the acute effects of outdoor green exercise different to exercising indoor exercise? (3) Are the acute effects of exercising indoors with virtual exposure to nature (‘virtual green’ exercise) different to indoor exercise? (4) Are the acute effects of outdoor green exercise different to virtual green exercise?

## 2. Materials and Methods

### 2.1. Search Strategy

The present review was not pre-registered, and represents an update of Thompson Coon et al. [18], whose searches were conducted in April 2010. We used the decision framework proposed by Garner and colleagues [25], and based our decision on the likelihood that there was sufficient number of new studies that could change the findings of the original review. We applied the same search strategy as Thompson Coon et al. to find eligible trials in the following databases: PubMed, CENTRAL, Embase (via OVID interface), and PsycINFO, GreenFile, Sports DISCUS (all via the EBSCO interface) (see Table 1 for our PubMed search strategy, and https://osf.io/mgfsd/ for all search strategies). We retained the eligible trials from Thompson Coon et al. [18], but screened their full texts for eligibility. All searches were run from 1st January 2010 to 28th June 2018. In addition to database searches, we also checked the references of included articles for any other eligible references.

### 2.2. Eligibility Screening and Data Extraction

We applied similar eligibility criteria as Thompson-Coon et al. [18], however we were more explicit regarding how we classified ‘green exercise’. Specifically, trials were eligible if the outdoor or virtual exercise conditions exposed participants to views of nature (see full eligibility criteria in Table 2). We applied no language restrictions, and contacted authors of potentially eligible trials where the information provided was insufficient to judge eligibility.

Two authors (I.M.L., P.D.) first screened the titles and abstracts of articles identified by the search strategy, and then retrieved and screened the full-texts of these articles. After full-text screening, the same authors (I.M.L., P.D.) extracted data from all trials that met our eligibility criteria. Data were extracted on country of origin, participant characteristics, intervention description and characteristics (i.e., exercise setting, type, duration, intensity, and frequency), environmental conditions, information regarding methodological quality, outcomes assessed, measurement tools used, statistical analysis details, and effects on outcomes (see Table 3 for study characteristics). Any disagreements during screening or data extraction were resolved by two the other authors (C.G., G.C.).

### 2.3. Assessment of Risk of Bias

Two authors (I.M.L. and P.D.) assessed risk of bias using the Cochrane “risk of bias” tool in the included RCTs. Specifically, we made judgements regarding the level of risk (low, high, or uncertain) for selection bias (allocation sequence generation and allocation concealment), performance bias (blinding of participants and personnel), detection bias (blinding of outcome assessors), attrition bias (incomplete outcome data), selective outcome reporting bias, and other bias (baseline imbalances and exercise adherence) (for a full description of this method see [26]). For crossover trials, we applied a modified “risk of bias” tool described by Ding and colleagues [27]. Briefly, this tool includes additional domains for potential bias, such as the appropriateness of the cross-over design, potential carry-over effects, and presentation of unbiased data, to the six RCT “risk of bias” domains described above. The two other review authors (C.G. and G.C.) arbitrated conflicts not due to assessor error.

### 2.4. Data Synthesis

We performed a meta-analysis on an outcome only if there were two or more studies that assessed that same outcome, and if outcome measures were not too diverse. We aimed to examine both post-intervention values and change from baseline score, however, too few trials included change scores to meta-analyze, hence, all analyses are based on post-intervention values.

For the meta-analysis, we used the inverse variance random-effects method [28] via RevMan software (Copenhagen, Norway) which uses Hedges adjusted g to calculate a standardized mean difference (SMD), adjusted for small sample bias. When trials measured an outcome using the same measurement method or scale we expressed the effects as mean difference (MD), whereas we used SMD when trials used different instruments or scales to measure the same outcome. We presented pooled intervention effect estimates and their 95% confidence intervals (CIs) for each outcome.

For trials that included more than one applicable green exercise or comparison group (e.g., [29]), we created (when possible) a single pair-wise comparison by combining outcome data (see [30]). If variability was presented by measures other than standard deviation (e.g., standard error, 95% CI, *t*-values, and *p*-values), we obtained an estimate of the standard deviation (SD) using standard approaches for transforming data via the inbuilt RevMan calculator [30].

Based on the recommendations of Elbourne et al. [31], we conducted a meta-analysis of crossover data only when trials provided results of paired analyses. For comparisons where meta-analysis was not possible and for outcomes where there were at least two trials available we produced harvest plots to summarize effects [32,33]. For all other outcomes we provided a narrative summary.

For each outcome, we assessed study heterogeneity using Cochran’s χ^2^ (Chi^2^) test [34], with *p* < 0.10 indicating evidence of heterogeneity, and I^2^ statistic, which describes the percentage of variability in point estimates that is due to heterogeneity rather than to sampling error [35]. In accordance with Higgins and Green [30], we interpreted I^2^ values as follows: 0%–40% as ‘might not be important’; 30%–60% as ‘may represent moderate heterogeneity’; 50%–90% as ‘may represent substantial heterogeneity’; 75%–100% as showing ‘considerable heterogeneity’.

## 3. Results

### 3.1. Search Results

We retained 10 of the 11 eligible trials in Thompson Coon et al. [18]—one trial [36] was excluded because the study design was ineligible. The updated search revealed 18 new trials from 2198 potentially eligible articles after removal of duplicates (see Figure 1). Therefore, by combining eligible trials from the original review and updated search, we included a total of 28 eligible trials from 31 papers in the current review. Most trials had a single publication (*n* = 25), whereas three trials had two publications each [37,38,39,40,41,42]. The characteristics of eligible trials can be found in Table 3; however, here we include a brief summary of the key characteristics.

### 3.2. Trial Design Characteristics

Most trials were acute (*n* = 25), and three were longitudinal RCTs [29,38,39]. However, one of the longitudinal studies [38] consisted of an intervention involving just two bouts of each exercise condition over a fortnight (with a 10-week follow-up assessment)—the remaining two studies were both 12 weeks in duration. Of the acute studies, nine were randomized crossover trial (RXT) design, eight were non-RXTs, seven were RCT parallel group studies, and one was a quasi-RCT parallel group design. The longitudinal trials were conducted in Canada [39], Iran [29], and Norway [38]. Of the 25 eligible acute trials, most were based in either North America (USA, *n* = 8, and Canada, *n* = 2) or the UK (*n* = 8). Characteristics of the individual studies are presented in Appendix A (see Appendix A), but we also provide a brief summary of these characteristics below.

### 3.3. Participant Characteristics

A total of 1344 (median: 34.5, min–max: 8–181) participants were recruited to the 28 eligible trials. The total sample of the three longitudinal RCTs was 112 (median: 31, min–max: 14–75), and the mean (SD) age was 51 (9) years. Samples in these longitudinal trials comprised of sedentary, postmenopausal women [39], Iranian women with severe depression, obesity, and vitamin D deficiency [29], and male and female (50% each) office workers [38].

In the 25 acute trials, a total of 1232 (median: 35, min–max: 8–181) participants were recruited, with a mean ± SD age of 30 ± 13 years (data from 19 trials). The samples in these acute trials comprised of university students (*n* = 10 studies), university students and staff (*n* = 2), recreational or competitive athletes (*n* = 5), healthy adults (*n* = 5), postmenopausal women (*n* = 2), patients post-stroke (*n* = 1), and primary school kids (*n* = 1). Most acute studies (*n* = 16) consisted of both male and female participants (mean % female vs. male: 55% vs. 45%), whereas four trials each comprised of female- [43,44,45,46] and male-only samples [14,47,48,49]—one trial [50] did not report participants gender.

### 3.4. Longitudinal Intervention Characteristics

All three eligible longitudinal RCTs compared only outdoor green exercise with indoor exercise [29,38,39]. One trial compared running outdoors with indoor treadmill running [29], whereas another [38] utilized a combined cycling (bicycling outdoor forest area vs. indoor static cycling) and strength training (outdoor grass yard vs. indoor) intervention. Similarly, Lacharité-Lemieux and colleagues [39] combined strength training with aerobic circuit training (outdoor natural park vs. indoor).

The intensity of exercise prescribed was moderate in two of these trials ([38], 55% heart rate (HR) reserve; [29], 55%–75% target heart rate, HR) and moderate-to-high in the other ([39], 65%–95% maximal HR). Exercise duration ranged from 45 min [38] to an hour [39]. Although only one longitudinal RCT [38] provided environmental conditions for outdoor (temperature: 8–10 °C) and indoor (20 °C) settings, the other two trials did give the time of year the intervention took place ([29], Fall; [39], April–July).

### 3.5. Acute Intervention Characteristics

#### 3.5.1. Mode and Settings of Outdoor Green Exercise

Of the 21 studies that involved outdoor green exercise, the chosen modes of exercise were walking (*n* = 14), running (*n* = 4), cycling (*n* = 2), and dancing (*n* = 1). Outdoor walks and runs took place on woodland trails [47,51,52] or footpaths through gardens, parks, or nature [43,45,48,50,53,54,55,56,57,58,59,60]. One trial compared mountain hiking in an Austrian Alpine region with treadmill walking, flat and on an incline [42], whereas another compared salsa dancing outdoors with indoor salsa dancing [61]. The cycling trials compared bicycling along a recreational trail [49] or on cycle ergometer located on a sports field surrounded by grassland and trees [62] with cycling on an indoor ergometer.

#### 3.5.2. Intensity, Duration, and Environmental Conditions of Outdoor Green Exercise

Exercise intensity was moderate in all but one study—Turner and Stevinson [52] instructed participants to run as fast as possible in the last 3 km of a 6 km run. Six trials set intensity using percentage of either heart rate (HR) reserve/maximum [43,44,47,62,63] or percentage of maximal oxygen uptake (VO_2max_, *n* = 1, [48]), whereas other studies reported either speed [41,52] or time taken to cover a specific distance [14,59] as an indication of intensity. Additionally, exercise intensity was described as self-paced at a ‘comfortable’ or ‘consistent’ pace in nine studies [45,49,51,53,54,55,56,57,58], and as moderate-to-vigorous in one trial [61]. Short duration exercise bouts ≤20 min were used in most studies; only three trials included exercise bouts of 40 minutes or more ([61], 40 min; [45], 1 h; [41], 3 h). Four studies used a set distance rather than prescribed exercise duration ([47], 5 km; [48], 10 miles; [49], 40 km; [52], 6 km).

In eight of the trials, interventions took place in summer to autumn; the remaining 17 studies did not report when the study took place. While 12 trials provided some details about the weather conditions during outdoor exercise, only five trials [45,48,49,52,53] provided environmental condition data from both outdoor and indoor conditions (temperatures slightly colder outdoors, mean: 21 vs. 23 °C).

#### 3.5.3. Mode and Settings of Indoor Virtual Green Exercise

In the nine trials that included a virtual green exercise condition, exercise consisted of cycle ergometer exercise (*n* = 2; [46,63]) or treadmill walking (*n* = 5; [53,55,56,57,58]) or running (*n* = 2; [64,65]). The virtual exercise condition in all of the five studies that involved treadmill walking consisted of watching a video (viewed either on a screen or with virtual reality technology) of the same route performed in the outdoor condition. In the remaining trials, participants watched videos of a trail through a forest area [63,64], fields, hedgerows, and woodland [46], and paths, gardens, trees, and flowerbeds [65].

#### 3.5.4. Intensity, Duration, and Environmental Conditions of Indoor Virtual Green Exercise

All of the virtual conditions included moderate-intensity ([63], 50% HR reserve; [57], “brisk walk”; [64], 60% VO_2max_; [46], 55% maximal HR) or ‘self’ or ‘comfortably’ paced exercise [53,55,56,57,58,65]. Rogerson and Barton [64], however, also included a high-intensity (85% VO2_max_) exercise condition. Exercise duration was either 10 [53,56], 15 [46,53,55,56,57,58,63,64], or 20 [57,58,65] minutes. No trial described the environmental conditions during exercise bouts.

#### 3.5.5. Comparison Conditions

Outdoor exercise was compared with indoor treadmill walking or running in a laboratory or fitness center in 15 studies [14,43,45,46,47,48,52,53,55,56,57,58,63,64], and with walking through indoor hallways/tunnels [50,54,59,60] or a shopping center [51]. Participants in one of the indoor conditions exercised while watching ‘self-selected entertainment’ [65].

### 3.6. Study Outcomes and Measures

Full details of the outcomes and measurement tools used can be found in Table 3. Of the three longitudinal RCTS that examined green exercise compared to indoor exercise, all included measurements for both subjective and objective outcomes [29,38,39]. Subjective outcomes included affect, depression, restoration, perceived effort, enjoyment, physical activity levels, and physical activity intention, of which most were measured using reliable and validated scales. Objective outcomes included biomarkers, cardiovascular responses, anthropometric measurements, cardiorespiratory fitness, exercise adherence, and muscular strength and endurance.

In the eligible acute trials, subjective outcomes included affect/mood, emotions (e.g., energy, tension, calmness, depression) enjoyment, self-esteem, perceived effort, physical activity intention, time perception, perceived restoration, and cyber sickness. Objective outcomes across eligible acute trials included attention and cognitive function, cardiovascular and physiological responses (e.g., heart rate and blood pressure), biological markers, power output, walking/running, speed, distance covered, and time to completion and exhaustion.

### 3.7. Risk of Bias

Authors’ judgements and rationales for risk of bias can be found in Appendix A. Below we briefly summarize the risk of bias for RCTs and crossover trials. Summary of judgements across trials are provided in Figure 2 and Figure 3, and for individual trials in Appendix A (see Appendix A).

#### 3.7.1. Risk of Bias in Randomized Comparative Trials

All but three of the 11 RCTs were at an unclear risk of selection bias due to inadequate information about randomization procedures provided in the papers. Two trials [38,39] used acceptable randomization procedures but did not conceal condition allocation from the trial personnel. One trial [55] was at a high risk of selection bias because it adopted a quasi-randomization method, whereby the outside temperature and the presence of rain influenced the order in which conditions were allocated.

Due to the inability to blind participants to the nature of both the intervention (exercise) and the environment (outdoor vs. indoor) we judged all trials at high risk of performance bias. Only one trial [39] stated that they blinded outcome assessors to the intervention—all other trials were judged at high risk of detection bias. All but one trial were deemed at low risk of attrition bias due to minimal or no loss to follow-up; in Byrka and Ryczko [61] 54% of participants did attend their assigned condition. Due to a lack of pre-registered reports or published protocols, all but two trials were judged to have an unclear risk of selective reporting. Two trials [38,55], however, did not report all of the items provided by the measurement tools they used and were, therefore, deemed at a high risk of selective reporting bias.

Two trials were judged at a high risk of other bias, one [38] because of a very small sample size (*n* = 7 per group) and a short duration of exercise intervention (2 exercise sessions over two weeks), and another [55] because the virtual green exercise stimulus was changed half-way through the experiment. One trial [50] was judged to have an unclear risk of other bias due to a lack of participant characteristic data in the report, and all other trials appeared to be free of other biases.

#### 3.7.2. Risk of Bias in Crossover Trials

In four [41,53,54,59] out of the 17 crossover trials we judged that the crossover design employed may not have been appropriate due to a short wash-out period between conditions (see Figure 3). Nine (53%) trials had an appropriate design, whereas there was insufficient information in four trials [14,48,49,52] to make a judgement. Similarly, due to insufficient wash-out periods, lack of evaluation of potential carry-over effects, or inadequate information, the same trials were judged at a high [41,53,54,59] or unclear [14,48,49,52] risk of a carry-over effect influencing study results. All trials were deemed to have included unbiased data because data from all treatment periods were provided.

The eight eligible non-RXTs were judged at a high risk of selection bias. Of the nine RXTs, seven were deemed to have an unclear risk of selection bias because the randomization method was not reported. Only two trials provided the randomization method used ([54], drawing numbers from a hat; [52], online random number generator). For all trials however, either allocation was not concealed (*n* = 10) or allocation concealment was unclear (*n* = 7). Again, because it is not possible to blind participants to exercise and the environment setting, we judged all trials at high risk of performance bias. Similarly, none of the trials stated that they blinded outcome assessors to the intervention, and thus were deemed to be at high risk of detection bias. Most trials (*n* = 14) reported minimal loss to follow-up. Three trials [41,52,65] were judged at high risk of attrition bias, because some participants (10%–20%) did not complete all conditions. All trials had an unclear risk of selective reporting due to a lack of pre-registration. Six trials were deemed at risk of other biases due to; baseline imbalances [54]; very low sample sizes (all *n* > 15, [14,49,63,64]); and concerns over the control of confounders ([49], wind speed and direction; [41], replication of mountain hiking on a treadmill).

### 3.8. Summary of Effects: Longitudinal Outdoor Green Exercise Interventions Versus Indoor Exercise

#### 3.8.1. Meta-Analysis Results

Meta-analysis was possible for positive affect/engagement, tranquility, depressive symptoms, rate of perceived exertion (RPE), average percentage heart rate during exercise, systolic and diastolic blood pressure (SBP and DBP), physical activity, body mass, body mass index (BMI), and percentage body fat. All trials reported post-intervention values (none provided mean changes). Each analysis involved data from just two trials, so caution is advised when interpreting the results of our meta-analysis. Of the analyses performed, slightly lower post-intervention RPE values with green exercise were the only statistical difference observed (Table 3). None of the longitudinal studies performed a prospective power analysis, therefore, it is unclear whether any were sufficiently powered to detect changes in the outcomes they assessed.

#### 3.8.2. Outcomes Not in the Meta-Analysis

Most of the outcomes assessed in the three longitudinal trials could not be included in a meta-analysis, so we report these outcomes qualitatively briefly here and in Appendix A (see Appendix A). Statistically favorable effects of green versus indoor exercise were found for perceived restorativeness of the environment, enjoyment, intention to exercise in the future (after correcting for previous exercise behavior), salivary cortisol awakening response (CAR) area under the curve with respect to increase [38], adherence to exercise [39], waist-to-hip ratio (in exercise groups only), and serum 25-hydroxyvitamin D concentrations [29].

One study [38] found no statistical between-condition differences in heart rate achieved during exercise, CAR area under the curve with respect to the ground, or serum cortisol concentration. Another study [39] found no effects on revitalization, fatigue, average and maximal heart rate achieved during sessions, estimated VO_2max_ (a pre-post increase in the indoor exercise group only), upper (bench press and lat pulldown) and lower (leg press) body muscular strength and endurance, blood glucose-insulin dynamics, lipids, waist circumference, or fat mass and lean body mass (derived via DXA) [39]. However, despite no statistical time by environment interaction for upper (bench press and lat pulldown) and lower (leg press) body muscular strength, Lacharite-Lemieux et al. The study in [39] did report statistically improved upper body muscular strength and endurance in the green exercise group only.

### 3.9. Summary of Effects: Acute Outdoor Green Exercise Versus Indoor Exercise Without Exposure to Nature

Twelve crossover trials [14,41,43,45,47,48,49,51,52,54,59,62] and five parallel RCTs [44,50,57,60,61] compared the effects of acute outdoor green exercise with indoor exercise. Only four of these trials were sufficiently powered to detect at least moderate effects on affective states [41,43], affective valence [52], and power output [49].

#### 3.9.1. General Mood or Affect

Nine trials assessed a range of general affect constructs with a variety of tools (Figure 4). All but three of the trials examined the acute effects of exercise bouts of 30 min or less, and only two ([61], salsa dancing; [62], stationary cycling) did not use walking or running as their exercise mode. The effect of green versus indoor exercise on general affect was inconclusive (Figure 4). However, five of the trials (2 RXTs, 1 non-RXT, and 2 RCTs) showed statistical improvements in general affect with green versus indoor exercise, whereas the other four trials (2 RXTs and 2 non-RXTs) showed no statistical effect. Effect sizes of three trials that reported these ranged between a moderate and large beneficial effect (d = 0.54 to 1.45).

Only three trials (all RXTs) included affective valence (assessed via feeling scale (FS)) and perceived activation (via felt arousal scale (FAS)) as outcomes (Figure 4). Two trials observed statistically higher affective valence scores and one trial found statistically improved perceived activation with green versus indoor exercise. The exercise used in these trials were diverse: a 10-min walk in nature [43], a 6-km outdoor run [52], and a 3-h mountain hike in an Alpine region [41]. Both Niedermeier et al. [41] and Focht et al. [43] were sufficiently powered to detect a medium effect of green exercise on affective states, whereas Turner et al. [52] was specifically powered to detect at least a 1-unit change in affective valence. It also should be noted that the trials [41,52] showing no effect of green exercise on affective valence and perceived activation employed Bonferroni corrections lowering their threshold for statistical significance to *p* < 0.025 and *p* < 0.0125, respectively.

#### 3.9.2. Emotions

When interpreting the results below it is important to note that only two trials were powered to detect at least medium changes in affective state outcomes [41,43]. Seven (3 RXTs, 2 RCTs, and 2 non-RXTs) of the ten trials that included a measure of ’energy’ found no difference between real green exercise and indoor exercise (Figure 4). Two randomized trials (1 RXT and 1 RCT) observed an increase in ‘energy’ after a 10–15 min walk in nature versus indoor treadmill walking, and one non-RXT found an effect with a 45 min run in nature.

Most trials (*n* = 4/6, 67%) observed no statistical effect of green exercise versus indoor exercise on measures related to calmness (i.e., relaxation and tranquility, see Figure 4). All but one trial was not randomized, and all trials had sample sizes above 35. The two trials [41,50] finding a statistical effect of green exercise reported small to large effects (d = 0.27 and 1.09). One of these trials consisted of a 3-h mountain hiking intervention [41].

Half (*n* = 4/8) of the trials that included a measure of anxiety/tension/worry found a favorable statistical effect of real green exercise versus indoor exercise (Figure 4). Four were crossover trials, three trials employed a non-randomized design and the other was randomized, and two trials had sample sizes below 20 participants. Of the four trials (2 RCTs, 1 RXT, and 1 non-RXT) that found no difference between outdoor green and indoor exercise, all exercise bouts were below 30 min and all sample sizes were at least 24 participants.

Out of the six trials (2 RXTs and 4 non-RXTs), only two non-RXTs [14,51] found a beneficial statistical effect of green exercise versus indoor exercise on anger (Figure 4). The same two trials also found a statistical improvement in depression (including sad and sullenness; Figure 4). Statistically lower fatigue scores (including physical exhaustion and tiredness) were observed with green versus indoor exercise in just three (1 RCT, 1 RXT, and 1 non-RXT) of the eight trials that included this outcome (Figure 4). However, in one of the RCTs this statistical effect was observed in female participants only [57]. Only one non-RXT [51] reported statistical improvements in confusion with green versus indoor exercise, whereas two trials (1 RXT [62] and 1 non-RXT [45]) reported no statistical difference between-conditions for confusion (not shown in figure). Compared with equivalent duration indoor treadmill walking, statistical improvements in happiness (including elation) was observed after a 3-h mountain hike [41], but not after a 60-min nature walk [45] (not shown in figure).

Three out of four trials found statistically higher enjoyment or satisfaction scores after real green exercise versus indoor exercise (Figure 4). The exercise interventions that found greater enjoyment with green exercise consisted of 10- [43] and 20-min walks [44], and a 12-km run [14]. Peacock et al. [51] measured enjoyment of exercise using a single-item question (0–5 scale), and reported slightly higher average scores after green exercise compared with indoor exercise (mean: 4.4 vs. 3.6, no statistical analysis reported). Another trial [59] asked participants whether they found a nature walk or an indoor walk as the “most beautiful” and “most enjoyable”; 11 of the participants chose the nature walk and 10 chose the indoor walk.

Some emotion outcomes were reported only by one trial each. In these trials, statistically favorable scores for pleased (β = 0.8, 95% CI 1.7, −0.1, *p* = 0.03), frustrated (β = 0.9, 95% CI 0.12, 1.6, *p* = 0.03) [45], self-esteem (*p* < 0.05; [51]), fascination, and nature relatedness scores (*p* < *0*.01, d = 1.13, *p* < 0.01, d = 0.86, respectively; [50]) with green exercise versus indoor exercise. No statistical effects, however, were found for delighted (β = 0.9, 95% CI 1.75, −0.02, *p* = 0.05), joy (β = 0.6, 95% CI 1.23, −0.05, *p* = 0.07) [45], contemplation (*p* = 0.272, d = 0.27; [41]), total pleasant somatic, total unpleasant, total unpleasant somatic emotions, and a range of other emotions (boredom, placidity, anger, provocativeness, humiliation, modesty, shame, gratitude, resentment, virtue, and guilt) with green exercise compared with indoor exercise [47]. However, Kerr and colleagues [47] found that tension and effort stress scores were statistically higher (*λ* = 0.82, *p* < 0.05) in competitive athletes performing green versus indoor exercise. Conversely, recreational runners after green exercise reported statistically greater feelings of pride (*p* < 0.05) than after indoor exercise.

#### 3.9.3. Intention for Future Exercise Behavior and Social Interaction Time

Focht [43] found statistically greater intention for future participation (*p* < 0.001; d = 0.92) following 10 min of outdoor walking versus indoor walking. Conversely, another trial [62] reported no statistical effects on intention for future exercise behavior with 15-min on a cycle ergometer outdoors versus indoors. Rogerson and colleagues [62] also assessed social interaction time (via direct observation) and found a statistical effect for condition (*p* < 0.001, ηp^2^= 0.67), with greater social interaction time in the green exercise condition compared with the indoor condition.

#### 3.9.4. Attention and Memory

Two trials (1 RXT and 1 non-RXT) reported statistical time by condition interaction effects for attentional focus (via in the 28-item checklist of words, [42], *p* < 0.001) and directed attention (via digit span backwards task, [62], *p* = 0.02; Figure 2). Mieras and colleagues [49], however, found comparable attentional focus (via Tammen attentional focus scale) after outdoor and laboratory cycling trials (*p* = 0.261). Similarly, another RXT [59] found no statistical effect of 10-min walks in nature, urban, or indoor environments on recall and recognition for word lists.

#### 3.9.5. Perceived Exertion, Cardiovascular Responses, and Exercise Performance

Six trials (5 RXTs and 1 non-RXT) that measured perceived exertion via Borg’s RPE scale (6–20) reported no statistical difference between conditions (Figure 4). Only one trial [14] that used their own 0-8 scale reported statistically lower perceived exertion scores (*p* < 0.005) with green versus indoor exercise with internal focus (sound of breathing), but not with an external focus (outdoor noises).

Two RXTs reported statistically higher heart rates during green exercise versus indoor exercise ([49], *p* < 0.05; [41], *p* = 0.001 and d = 0.59), but two other trials [43,48] found no differences between conditions. The higher heart rates observed in Mieras et al. [49] are probably explained by higher power outputs and skin temperature. Whereas in Niedermeier et al. [41], when average heart rates were expressed as a percentage of maximum heart rate (% HRmax), the difference between conditions was minimal (mean: 59% vs. 57%). One trial [48] also found similar oxygen uptake and beta-endorphin responses, but statistically higher average blood lactate values after a 10-mile outdoor run compared with an indoor treadmill run (mean: 4.1 vs. 1.8 mmol/L). Two crossover trials [14,41] found no statistical differences in blood pressure responses with green versus indoor exercise. Niedermeier and colleagues [41] also observed no statistical differences in heart rate variability between mountain hiking and treadmill walking conditions.

Three crossover trials [45,49,52] and one parallel RCT [54] included exercise performance outcomes. One trial [49], powered to detect power output differences between outdoor and indoor cycling, found statistically higher power output and lower time to completion (*p*’s < 0.001) in the outdoor trial compared with indoors. Conversely, Turner and Stevinson [52] found only a 12 s difference in completion times of a maximum paced 3000-m run between an outdoor green and an indoor gym setting (*p* = 0.64). Another trial [45] also observed similar average self-chosen walking speeds during a 1-h walk outdoors versus indoors. Carvalho and colleagues [54] observed no difference in speed during a 30-m walk performed outdoors compared with indoors in patients recovering from stroke, but did find that those patients with a self-selected walking speed of ≥ 0.8m/s covered statistically greater distances in a six minute walk test performed outdoors compared with indoors (*p* = 0.01).

#### 3.9.6. Biological Markers

A non-RXT [14] reported statistically higher cortisol and noradrenaline (but not adrenalin) concentrations after an indoor run with internal attention focus than a run outdoors. Another trial [41], however, found no statistical interaction between outdoor mountain hiking and indoor treadmill walking for salivary cortisol, but noted comparable decreases after both conditions. Finally, Teas and colleagues [45] revealed statistical associations between stress hormones (salivary cortisol and log alpha amylase) and some emotions (sad, frustration, delight, pleased, and not anger), after controlling for exercise setting, but did not compare stress hormones concentrations in different exercise settings (outdoor vs. indoor).

### 3.10. Summary of Effects: Acute Effects of Virtual Green Exercise Versus Indoor Exercise Without Exposure to Nature (i.e., Non-Virtual)

Four crossover trials [46,63,64,65] and one RCT [57] compared the effects of acute virtual green exercise conditions with other non-virtual indoor exercise conditions (blank/neutral screen, [46,63,64]; “no visual stimulus”, [57]; virtual built environment, [46,64]; virtual blue condition: [46]; static virtual green, dynamic virtual green, and self-selected entertainment conditions, [65]). All five trials included exercise bouts of 15- or 20-min duration, and all but one of the trials [57] had relatively small sample sizes (≤ 30). None of these studies performed a prospective power analysis.

#### 3.10.1. General Affect

Only one RXT [46] included affective valence (via FS) and perceived activation (via FAS) as outcomes, and revealed statistically higher affective valence scores pre to 5-min during exercise for virtual green exercise compared with indoor cycling viewing a blank wall (*p* < 0.001). However, no statistical time by condition interaction was observed for perceived activation scores (via FAS).

#### 3.10.2. Emotions

In comparison, including one RCT [57] and two non-RXTs [63,65], no statistical differences were found between virtual green exercise and non-virtual indoor exercise for energy (including excitement and vigor), fatigue/tiredness, calmness, and anger (Figure 5). Two trials [63,65] found no difference in tension between virtual green exercise and non-virtual exercise. One trial [57], found statistically lower tension in female participants only after non-virtual exercise compared with virtual green exercise. Of the other emotions, Yeh and colleagues [65] found participants were happier in groups exercising while exposed to either a dynamic nature image and a static nature image compared with self-selected entertainment (*p* < 0.05).

#### 3.10.3. Liking the Activity and Likelihood to Repeat Exercise Behavior

One trial [46] reported a subjective evaluation of the activity using a single item question (0–6 scale) for ‘liking the activity’ and ‘likelihood to repeat exercise’. The authors [46] observed that virtual green exercise was evaluated more positively than exercising viewing a blank wall (*p* < 0.001), and participants reported they would be more likely to repeat the virtual green exercise (*p* < 0.001).

#### 3.10.4. Attention and Time Perception

In the one trial [64] that assessed directed attention (via backward digit span scores), improvements in attention were found for virtual green exercise (*p* < 0.001), but not for non-virtual exercise. Another trial [46] observed no statistical differences in time perception between virtual green and indoor blank screen exercise.

#### 3.10.5. Perceived Exertion, Exercise Intensity, Cardiovascular Responses, and Exercise Performance

Two RXTs [64,65] found no statistical differences in perceived exertion scores (via Borg’s RPE 6-20 scale) between 15 min of virtual green exercise and non-virtual indoor exercise conditions (Figure 5). One non-RXT [65] found that participants had statistically higher heart rates when exercising with self-selected entertainment than with exposure to a dynamic nature image and a static nature image (*p* < 0.05), whereas three trials (2 RXTs and 1 non-RXT) observed no statistical differences in heart rate between virtual green and indoor exercise (Figure 5). Similarly, another RXT [64] found similar energy expenditure and respiratory exchange ratio values between virtual green and indoor exercise conditions.

Two crossover trials (1 RXT and 1 non-RXT) measured blood pressure responses to virtual green versus indoor exercise. Duncan and co-workers [63] reported that SBP 15-min post-exercise was statistically lower after virtual green cycling exercise than cycling while viewing a blank screen (*p* = 0.01). No statistical differences were observed between the conditions for DBP. Similarly, White et al. [46] reported no statistical time by environment interaction for mean arterial pressure.

Two trials (1 RXT and 1 non-RXT) included exercise performance outcomes. The non-RXT [65] reported statistically greater distance run in the self-selected entertainment compared with the static nature image condition (*p* < 0.05), but not the dynamic nature image. Contrastingly, the RXT [64] reported no statistical between-conditions differences in time to exhaustion (*p* = 0.203).

### 3.11. Summary of Effects: Acute Effects of Outdoor Green Exercise Versus Indoor Virtual Green Exercise

One acute non-RXT [53] and four acute parallel RCTs compared green outdoor exercise with indoor virtual green exercise [55,56,57,58]. The sample sizes in these trials ranged between 26 and 181 participants, with three trials [55,57,58] consisting of samples above 112 participants. All five trials investigated the effects of relatively brief exercise bouts between 10 and 20 min duration. The five trials included here all lacked a prospective power analysis.

#### 3.11.1. General Affect

Three trials (2 RCTs and 1 non-RXT) included a measure of positive affect (Figure 6) and two trials (1 RCT and 1 non-RXT) assessed negative affect as an outcome (Figure 6). Only the non-RXT [53] found statistical improvements in positive affect with outdoor green versus virtual green exercise, and another trial [55] found no statistical differences between exercise conditions. Conversely, Gatersleben and Andrews [56] found that virtual nature walks performed after a mentally fatiguing task (Stroop test) were statistically more restorative for positive affect than outdoor nature walks (*p* < 0.01, d = 0.68). The authors also reported that no differences in positive affect were observed between walks that were high prospect-low refuge (i.e., clear lines of vision and few hiding places) compared with low prospect-high refuge (i.e., no clear lines of vision and many hiding places). One non-RXT [53] reported a statistical improvement in negative affect after outdoor versus virtual green exercise, whereas a RCT [55] observed no difference between groups (Figure 6).

#### 3.11.2. Emotions

Of the four trials (all RCTs) that included a measure of energy (including attentiveness), two trials [57,58] reported statistical increases in energy with outdoor green versus virtual green exercise, whereas the other trials [55,56] observed no between-condition differences (Figure 6). One of these trials [57] reported statistical improvements in energy with outdoor green exercise only in female participants. Gatersleben and Andrews [56] also noted that when analyzed by walk type, both the outdoor and indoor high prospect-low refuge walks resulted in statistical increases in attentiveness (*p*’s < 0.01).

Two trials [53,57] observed statistically higher feelings of calmness/tranquility after outdoor green exercise compared with virtual green exercise. Again, Plante and colleagues [57] found this statistical effect only in female participants. In contrast, Plante et al. [58] observed statistically higher calmness scores after a 10-min virtual green walk than with an outdoor walk. Only two trials (both RCTs) investigated the effects of outdoor versus virtual green exercise on tension [57,58]. One trial reported no statistical differences between exercise conditions, whereas Plante et al. [58], similar to the findings for calmness, observed statistically lower feelings of tension with virtual green versus outdoor exercise (Figure 6).

Gatersleben and Andrews [56] also observed that compared with virtual green exercise, outdoor green exercise resulted in statistical reductions in anger/aggression (*p* < 0.001, d = 1.07), but no effects on sadness, or fear arousal. Statistical interaction effects were also found between walk type and environment for attentiveness (*p* < 0.001), sadness (*p* < 0.001), fear arousal (*p* < 0.03), and anger/aggression (*p* < 0.03). Both virtual and outdoor high prospect-low refuge walks resulted in statistical increases in attentiveness (*p*’s < 0.01), and reductions in fear arousal (*p*’s < 0.001) and anger/aggression (*p*’s < 0.001). In contrast, low prospect-high refuge walks resulted in statistical increases in sadness (all *p*’s < 0.03), and only outdoor walks in low prospect-high refuge settings led to statistical increases in fear arousal (*p* < 0.001).

Three trials (2 non-RXTs and 1 RCT) that compared outdoor green versus virtual green exercise assessed enjoyment, but one trial did not perform a statistical analysis of this outcome. In that RCT [57], mean enjoyment scores were highest with outdoor green exercise and lowest with treadmill walking with virtual reality. In one non-RXT [53], enjoyment was statistically higher in the outdoor walk group than a virtual nature treadmill walk (*p* < 0.01), despite similar between-condition ratings of perceived environmental restorativeness (*p*’s > 0.05). The authors [53] attributed the lower enjoyment in the virtual nature condition to participant reported virtual reality-induced discomforts such as flatness, movement lag, and cyber sickness. The remaining RCT [58] reported statistically higher enjoyment scores (*p* < 0.05) in a group that walked briskly around a college campus than participants who walked indoors. The authors also found that female participants reported statistically higher enjoyment in the outdoor exercise condition than males (*p* < 0.05).

#### 3.11.3. Attention

Two RCTs assessed attention as an outcome. Gatersleben and Andrews [56] observed statistically greater attention scores (via Necker Cube Pattern Control task, NCPCT), with outdoor nature walks versus virtual nature walks (*p* < 0.001, d = 1.77). Statistical improvements in attention were observed in the high prospect-low refuge outdoor nature condition compared with the other three conditions (*p*’s < 0.01). Fuegen and Breitenbecher [55], however, in a much larger trial did not find statistical main effects of activity or environment, or activity by environment interaction effects for attention assessed via both the digit span backward and symbol digit modalities tests.

#### 3.11.4. Perceived exertion, exercise intensity, exercise performance

One non-RXT [53] assessed perceived exertion (RPE 6–20 scale), and observed statistically higher RPE during the virtual green walk than the outdoor walk (*p* < 0.001), despite no differences in walking speed (*p* = 0.07) or heart rate between the conditions. Two trials (1 RCT and 1 non-RXT) investigated the effects of outdoor green versus virtual green exercise on heart rate. The non-RXT [53] reported no statistical differences between outdoor nature walks and indoor virtual nature walks for average heart rate and highest heart rate achieved during the walks. Gatersleben and Andrews [56], however, observed that virtual nature walks resulted in statistically more restorative effects on heart rate than the outdoor walks (*p* < 0.001, d = 0.94).

## 4. Discussion

### 4.1. Summary of Findings

This current review updates and expands upon Thompson Coon et al.’s [18] systematic review that investigated the potential added health benefits of green exercise compared with indoor exercise. We identified 28 eligible trials—18 trials were found from the updated search and 10 were retained from Thompson Coon et al. [18]. Across the 28 eligible studies, we found largely inconclusive evidence for the benefits of outdoor and virtual green exercise over indoor non-green exercise. In our meta-analysis of three RCTs investigating the longitudinal effects of green versus indoor exercise, the only statistical effect was slightly lower post-intervention perceived exertion scores with green exercise. It was difficult to interpret the outcomes assessed in the longitudinal trials because most outcomes were assessed by single studies.

Of the affect and emotion outcomes assessed across 17 acute trials that compared outdoor green exercise with indoor exercise (i.e., without exposure to nature), only affective valence appeared to be more favorably affected by green exercise—although the number of trials with this outcome was small. More studies (3 out of 4) reported greater enjoyment or satisfaction after green versus indoor exercise. There were, however, consistent null findings (6 out of 7 trials) for the effect of green versus indoor exercise on perceived exertion, and equivocal findings for the effects on energy, calmness, tension, anger, depressed mood, fatigue, attention and memory, intention for future exercise behavior, biological markers, and exercise intensity and performance (walking/running speed, and heart rate). Compared with indoor exercise without exposure to nature, we found no consistent statistical effects on general affect, energy, tension, fatigue, perceived exertion, heart rate, or blood pressure with virtual green exercise. Across the five studies that compared the acute effects of outdoor green exercise with indoor virtual green exercise, no consistent differences were found between conditions for energy, calmness, tension, fatigue, attention, and heart rate response, however the two studies that included enjoyment as a measure reported statistically higher enjoyment scores with outdoor versus indoor green exercise.

### 4.2. Overall Completeness and Applicability of Evidence

There is a dearth of well-designed studies investigating the long-term effects of exercising with exposure to nature—we found only three that met our eligibility criteria and one of these was just two weeks duration. Whereas the majority of studies were conducted in North America and the UK, there was also representation from Scandinavia, Japan, Iran, Australia, and Central and Eastern Europe. There was an overrepresentation of young University-aged participants, but a fairly even gender split, in the acute studies reviewed. Most of the studies consisted of small sample sizes, with a median size of 33 (minimum-maximum = 8–181).

All acute studies used only one single episode of exercise per condition. Therefore, repeated bout effects are unknown. Walking and running were the most common exercise type, and woodland trails and footpaths through gardens and parks were the most common green exercise settings. However, the green exercise setting was not always well described. Similarly, authors did not report the environmental conditions consistently across the studies, which limits the generalizability of findings to different climates. Almost all studies consisted of moderate-intensity or “comfortable” self-paced exercise performed for short durations (most were ≤ 20 min). Most of the indoor comparison groups were performed on a treadmill, with only five studies including an indoor walking condition (i.e., walking through indoor hallways/tunnels/shopping center).

### 4.3. Quality of the Evidence

The current review includes evidence from 28 trials consisting of 1344 participants. However, we identified a number of methodological issues that resulted in an overall low quality of evidence. Imprecision resulted from small sample sizes and too few trials measuring the same outcomes with the same measurement tools. There was a high risk of bias across studies through: unclear randomization procedures; lack of allocation concealment; non-blinding of outcome assessors; scarcity of preregistration to rule out selective reporting; insufficient, unclear, or different washout periods in trials with crossover design (the effects of one environmental exposure might have influenced responses to the subsequent environment); and potential contamination in control conditions (e.g., possible exposure to green environments on way to indoor facilities). Most trials had insufficient statistical power (only 4 trials performed a priori power calculation), and a combination of low power, small samples, and lack of controlling for multiple comparisons (only three trials corrected for this) means the prevalence of false positives and false negatives is likely to be high. Inconsistency was evident in the diversity of ‘green’ setting, environmental conditions, exercise dose, control (or lack of) of exercise intensity, and outcome measures used. Poor reporting hampered both the extraction of methods information, outcome data, and risk of bias assessments.

### 4.4. Recommendations for Future Research

The fact that this review found largely inconclusive evidence for the benefits of outdoor and virtual green exercise over indoor exercise should not be interpreted as an appeal to researchers to stop conducting investigations in this area. We rather intend to set a challenge to researchers in this area to conduct more robust and rigorously designed trials to evaluate the effects of exposure to nature during exercise compared with exercise indoors. In particular, there is a clear need for trials with large enough samples to achieve sufficient statistical power, adequately report trials through the use of reporting guidelines (e.g., CONSORT), and to improve methodological transparency and rigor via preregistration of study designs and statistical plans, providing open data, appropriate randomization, allocation concealment, blinding procedures, and sufficient wash-out periods. Moreover, future robustly designed studies are needed to evaluate the effects of long-term exposure to nature during exercise and multiple bouts of longer duration exercise in nature in samples that are representative of the general population or in specific groups (e.g., people with depression or anxiety).

### 4.5. Biases in Review Process

We attempted to avoid bias by identifying all relevant studies through a comprehensive systematic search of seven major databases, screening all eligible studies for potential papers, and not excluding based on language. However, given the preponderance of small sample studies in this area there is a risk of publication bias—there are potentially “file drawers” with more null-findings from green exercise studies. We did not include or search the grey literature for unpublished studies or studies only published in an abstract form as these tend be of poor methodological quality, would not have been peer reviewed, and are often not reported well enough to extract sufficient information from [66,67].

Due to the small number of trials reporting on the same outcome, meta-analysis was possible for few outcomes. Instead for most outcomes, we used harvest plots to summarize and present data. Harvest plots are an extension of vote counting, whereby positive and null findings for each outcome using a predefined p-value threshold are counted [68]. Vote counting methods have been criticized because they do not provide measures of effect magnitude or take into account the size of the studies—a study with low power is less likely to find a statistically “significant” result even if there is a true effect. Therefore, by counting statistical and null-findings, we are potentially counting both false positives and negatives [69]. However, in the absence of a suitable alternative, the harvest plots allowed us to graphically display complex and diverse data (i.e., trial design, exercise bout duration, sample size, and p-values and effects sizes) within an easily interpreted format.

### 4.6. Agreements and Disagreements with Other Studies or Reviews

The equivocal findings in the current review is in slight contrast with a previous meta-analysis [16], which found more statistically favorable feelings of energy, anxiety, anger, fatigue, and sadness after direct exposure to a natural versus synthetic environment. In addition to the inclusion of a greater number of more recent trials, there are three important differences between the current review and Bowler and colleagues’ review [16], which makes comparisons between the two reviews difficult. Firstly, in addition to green exercise conditions, Bowler et al. [16] included non-exercise conditions that involved exposure to a natural environment while remaining passive or sedentary. Secondly, Bowler and colleagues’ [16] ‘synthetic environment’ conditions included built outdoor environments as well as indoor environments. Finally, Bowler et al. [16] performed a meta-analysis of outcomes, whereas, in the current review, in agreement with Thompson Coon et al. [18], this approach was deemed inappropriate due to a paucity of reported paired analysis. In their meta-analysis, Bowler et al. [16] did not take trial design into consideration, treating crossover trials as comparative trials (i.e., assuming treatment arms were independent), and combined both trial designs in their meta-analysis. This is the least desirable approach to combining crossover trials, and can overestimate the variability of the within-study treatment effect [70].

In their 2011 systematic review, Thompson Coon et al. [18] found favorable effects on feelings of anger, confusion, depression, energy, enjoyment and satisfaction, positive engagement, revitalization, and tension, but negative effects on feelings of calmness, with green exercise compared with exercising indoors. Similar to the current review, Thompson Coon and colleagues [18] also identified a number of issues with the available evidence, and concluded that the interpretation and generalization of findings was impeded by poor methodological quality and the diversity of outcome measures assessed in the eligible studies.

## 5. Conclusions

After a comprehensive review of the green exercise literature, we have failed to find support for the hypothesis that exercising in outdoor or virtual green environments offers superior benefits to exercising indoors without exposure to nature. We found tentative evidence that participants reported greater affective valence and enjoyment when exercising exposed to nature versus indoors, but that green exercise does not result in differential effects on affect, emotions, exercise intentions, attention, rate of perceived exertion, exercise intensity and performance, or biomarkers. This may be because there is no true additional benefit, or that the quality of the studies was inadequate to find a supplementary effect of green exercise. Future robustly designed studies are needed to evaluate the effects of acute and long-term exposure to nature during exercise in samples that are representative of the general population, as well as in specific groups.

## Figures and Tables

**Figure 1 ijerph-16-01352-f001:**
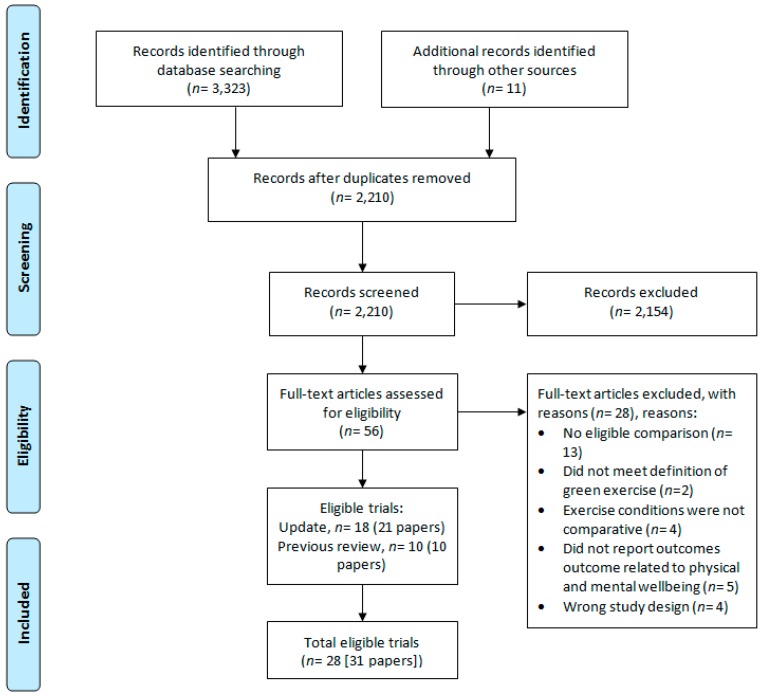
Flow of studies

**Figure 2 ijerph-16-01352-f002:**
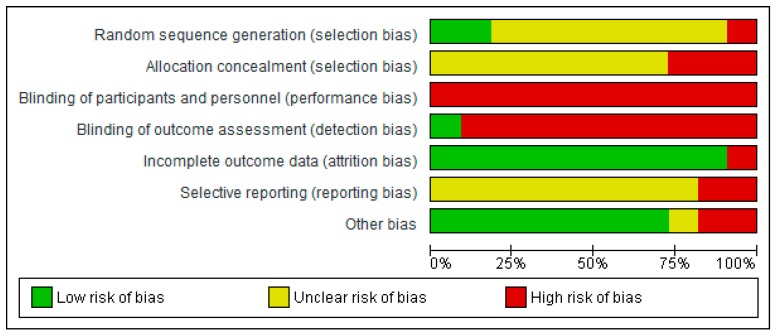
Risk of bias summary across randomized comparative trials: review authors’ judgements about each risk of bias item for each included study.

**Figure 3 ijerph-16-01352-f003:**
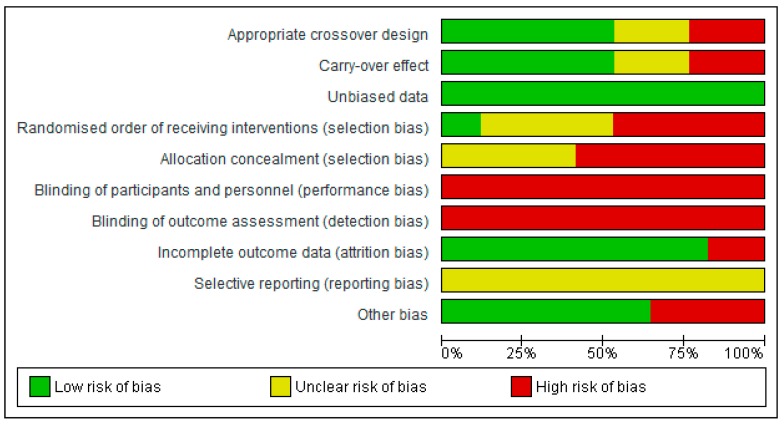
Risk of bias summary across acute crossover trials: review authors’ judgements about each risk of bias item for each included study.

**Figure 4 ijerph-16-01352-f004:**
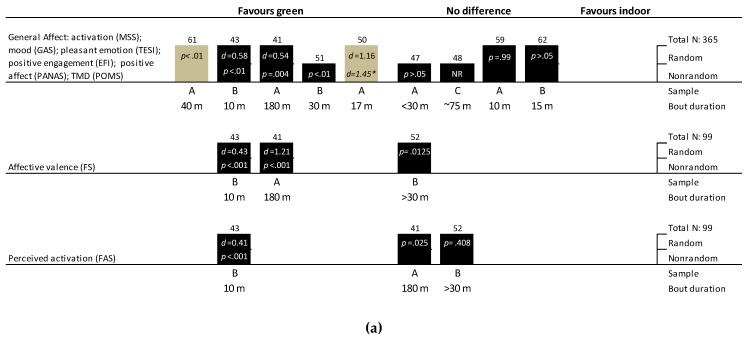
Harvest plots of trials investigating the effects of outdoor green exercise versus indoor green exercise without exposure to nature on (a) general affect and (b) emotions and perceived exertion. Note: A: studies with ≥ 40 participants; B: studies with 20–39 patients; C: studies with < 19 patients. We also provide the total sample across trials for each comparison. The taller bars represent randomized trials in this comparison, and shorter bars represent non-randomized trials. Grey bars indicate parallel group design trials, whereas black bars represent crossover design trials. *p*-values and effect sizes (i.e., Cohen’s d) are provided inside each bar where this information is available. Exercise bout duration is provided in minutes (Kerr et al. 2006 [47] consisted of 5-km runs which we characterized as less than 30 min; whereas Turner et al. 2017 [52] included 6-km runs which we characterized as > 30 min). The number on top of each bar represents each study’s reference: 14 = Harte & Eifert 1995 [14]; 41 = Niedermeier 2017 [41]; 43 = Focht 2009 [43]; 44 = Plante 2007 [44]; 45 = Teas 2007 [45]; 47 = Kerr 2006 [47]; 48 = McMurray 1998 [48]; 49 = Mieras 2014 [49]; 50 = Nisbet 2011 [50]; 51 = Peacock 2007 [51]; 52 = Turner & Stevenison 2017 [52]; 57 = Plante 2003 [57]; 59 = Rider & Bodner 2016 [59]; 60 = Ryan 2010 [60]; 61 = Byrka & Ryczko 2018 [61]; 62 = Rogerson 2016 [62]. Acronyms: AD-ACL: Activation–Deactivation Adjective Check List; BDST: Backwards Digit Span Test; EFI: Exercise-Induced Feeling Inventory; FS: Feeling Scale; FAS: Felt Arousal Scale; GAS: General affect scale; MSS: Mood Scale Score; NAS: Negative Affect Scale; PACES: Physical Activity Enjoyment Scale; PANAS: Positive and Negative Affect Schedule; PAS: Positive Affect Scale; POMS: Profile Of Mood States; STAI: State-Trait Anxiety Inventory; SVS: Subjective Vitality Scale; TAF: Tammen Attentional Focus; TESI: Tension and Effort Stress Inventory; VAS: Visual Analogue Scale; TMD: Total Mood Disturbance. * Internal replication study.

**Figure 5 ijerph-16-01352-f005:**
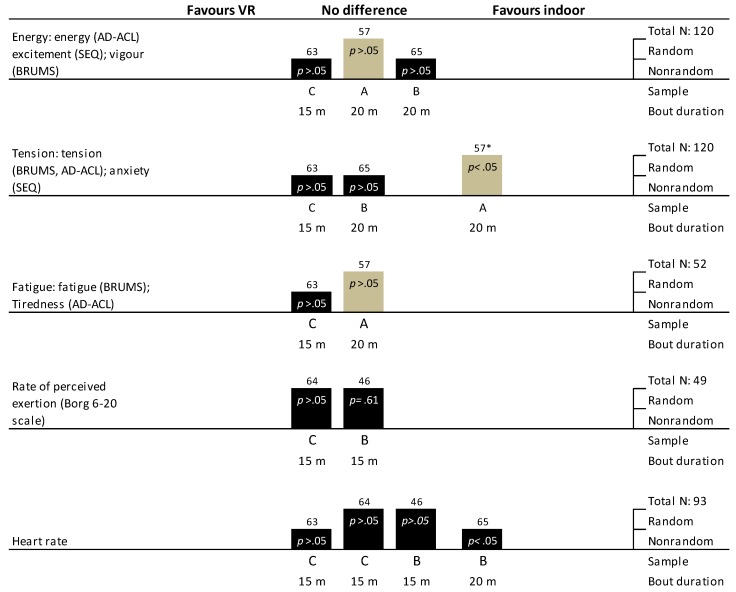
Harvest plots of trials investigating outdoor green exercise versus virtual green exercise. Note: A: studies with ≥ 40 participants; B: studies with 20–39 patients; C: studies with < 19 patients. We also provide the total sample across trials for each comparison (note: Plante et al. 2003 [57] did not provide group sample sizes, so we divided the total sample by the number of conditions to approximate group sizes). The taller bars represent randomized trials in this comparison, and shorter bars represent non-randomized trials. Grey bars indicate parallel group design trials, whereas black bars represent crossover design trials. *p*-values and effect sizes (i.e., Cohen’s d) are provided inside each bar where this information is available. Exercise bout duration is provided in minutes. The number on top of each bar represents the study: 46 = White 2015 [46]; 57 = Plante 2003 [57]; 63 = Duncan 2014 [63]; 64 = Rogerson 2015 [64]; 65 = Yeh 2017 [65]. Acronyms: VR: virtual reality; AD-ACL: Activation–Deactivation Adjective Check List; BRUMS: Brunel Mood Scale; SEQ: Sport Emotion Questionnaire. * Female participants only.

**Figure 6 ijerph-16-01352-f006:**
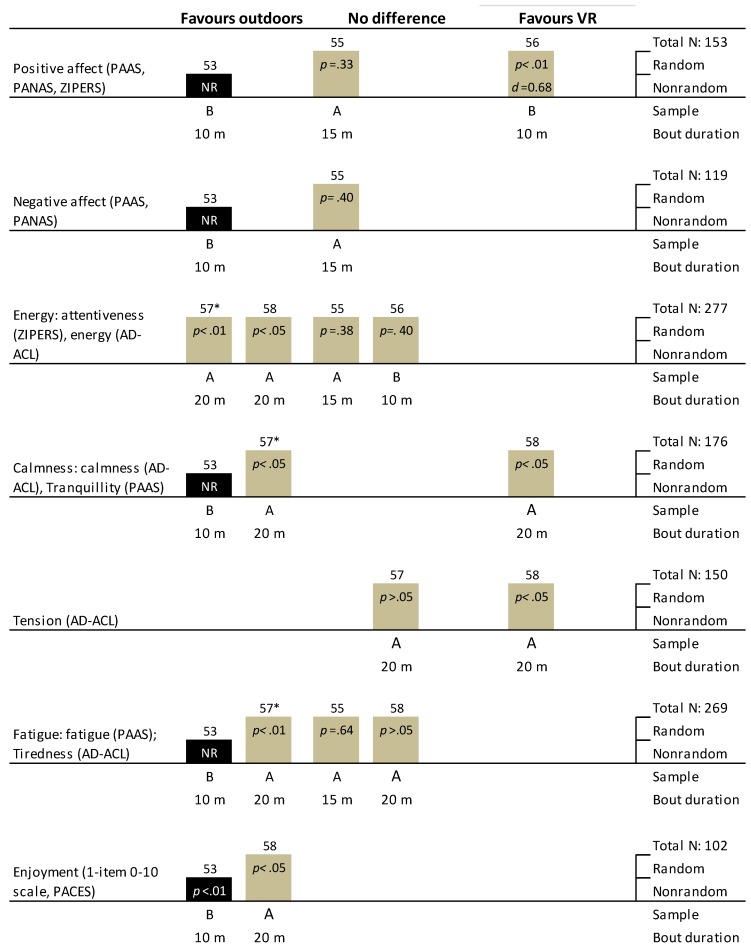
Harvest plots of trials investigating outdoor green exercise versus virtual green exercise. Note: A: studies with ≥ 40 participants; B: studies with 20–39 patients; C: studies with < 19 patients. We also provide the total sample across trials for each comparison (note: Plante et al. 2003 and 2006 [57,58] did not provide group sample sizes, so we divided the total sample by the number of conditions to approximate group sizes). The taller bars represent randomized trials in this comparison, and shorter bars represent non-randomized trials. Tan bars indicate parallel group design trials, whereas black bars represent crossover design trials. p-values and effect sizes (i.e., Cohen’s d) are provided inside each bar where this information is available. The number on top of each bar represents the study’s reference number in the present systematic review. Exercise bout duration is provided in minutes. The number on top of each bar represents the study: 53 = Calogiuri 2018 [53]; 55 = Fuegen & Breitenbecher [55]; 56 = Gatersleben & Andrews [56]; 57 = Plante 2003 [57]; 58 = Plante 2006 [58]. Acronyms: VR: virtual reality; AD-ACL: Activation–Deactivation Adjective Check List; PACES: Physical Activity Enjoyment Scale; PAAS: Physical Activity Affect Scale; PANAS: Positive and Negative Affect Schedule; ZIPERS: Zuckerman Inventory of Personal Reactions. * Female participants only.

**Table 1 ijerph-16-01352-t001:** Search strategy for PubMed (adapted for the other databases).

Number	Search Strategy
#1.	green exercis*.tiab
#2.	green gym*.tiab
#3.	ecotherapy.tiab
#4.	(outdoor* or outside*).tiab
#5.	(exercis* or physical activit* or walk* or physical fit*).tiab
#6.	#4 and #5
#7.	park*.tiab
#8.	#5 and #7
#9.	(greenspace* or green space*).tiab
#10.	#5 and #9
#11.	natural environment*.tiab
#12.	#5 and #11
#13.	nature.tiab
#14.	#5 and #13
#15.	(indoor or inside or laboratory or gym*).tiab
#16.	#1 or #2 or #3 or #6 or #8 or #10 or #12 or #14
#17.	#15 and #16

**Table 2 ijerph-16-01352-t002:** Green exercise review eligibility criteria.

PICOS	Inclusion Criteria
**Population**	Adults or children
**Interventions**	Studies must include experimental conditions in which participants *were explicitly/purposefully* exposed to views of nature (sceneries containing elements of nature such as trees, plants, grass, mountains, water, etc.) whilst engaging in exercise. The nature exposure could be achieved by having the participants exercising in *outdoor environments containing nature elements* or by exposing them to virtual sceneries of nature (e.g., images or videos of nature projected on a screen or viewed using virtual reality goggles).
**Comparison**	Exercise initiatives conducted indoors with no exposure to nature. The exercise must be of the same volume, duration, intensity, and mode as in the green exercise condition.
**Outcomes**	Any outcome related to physical and mental wellbeing.
**Study design**	Randomized crossover (RXTs) or controlled/comparative trials (RCTs), quasi-RXTs and quasi-RCTS, or non-RXTs and non-RCTs (both acute and longitudinal trials were considered)

**Table 3 ijerph-16-01352-t003:** The longitudinal effects of outdoor green exercise versus indoor exercise on emotions and enjoyment: meta-analysis of parallel RCTs.

Outcome	Trials	Sample	Statistical Method	Effect Estimate	I^2^
Positive affect/engagement	2	51	Std. Mean Difference (IV, Random, 95% CI)	0.94 [−0.59, 2.46]	84%
Tranquility	2	37	Std. Mean Difference (IV, Random, 95% CI)	0.25 [−0.40, 0.90]	0%
Depressive symptoms	2	83	Std. Mean Difference (IV, Random, 95% CI)	−0.58 [−1.81, 0.64]	84%
RPE	2	37	Mean Difference (IV, Random, 95% CI)	−1.02 [−1.88, −0.16]	0%
Average HR (% HR_max_)	2	37	Mean Difference (IV, Random, 95% CI)	−0.76 [−4.66, 3.14]	0%
Systolic blood pressure (mmHg)	2	37	Mean Difference (IV, Random, 95% CI)	3.39 [−2.80, 9.58]	0%
Diastolic blood pressure (mmHg)	2	37	Mean Difference (IV, Random, 95% CI)	−1.83 [−9.49, 9.55]	70%
Physical activity	2	37	Std. Mean Difference (IV, Random, 95% CI)	1.36 [−0.50, 3.22]	79%
Mass (Kg)	2	83	Mean Difference (IV, Random, 95% CI)	−0.15 [−2.10, 1.80]	0%
BMI (Kg/m^2^)	2	83	Mean Difference (IV, Random, 95% CI)	−0.10 [−1.01, 0.80]	0%
Body fat (%)	2	83	Mean Difference (IV, Random, 95% CI)	−1.43 [−5.12, 2.27]	63%

Note: IV, Inverse Variance; CI, Confidence Interval; SMD, standardized mean difference; MD, mean difference; RPE, rate of perceived exertion; HR: heart rate; BMI: body mass index.

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
