# Peer review of "The Effects of Green Exercise on Physical and Mental Wellbeing: A Systematic Review"

_ijerph, 2019, doi:10.3390/ijerph16081352_

Round 1

Reviewer 1 Report

This is a useful addition to the literature, and the authors do a good job of justifying the need for this review.

I am not an expert in systematic review and meta-analysis methods, but the design and reporting of the methods for this study are appropriate and described in good detail. The authors have done well to try to contain the research and limit the wide range of variables that exist in this area of the literature. I think that there are opportunities to identify relevant implications for this work, as these sorts of meta analyses are likely to be useful in driving health advice given by GP’s, and to drive policy change around the provision of outdoor exercise environments. My fear is that researchers will not continue to conduct research in this area given that this review and others have generally found inconclusive or lukewarm support for green exercise.

Overall, the article was exceptionally well written, with just a very minor suggestion. On Page 2 line 77- I would normally use ‘associated with’ or ‘linked to’ the notion of…

Author Response

Response to Reviewer 1 Comments

Point 1: This is a useful addition to the literature, and the authors do a good job of justifying the need for this review.

I am not an expert in systematic review and meta-analysis methods, but the design and reporting of the methods for this study are appropriate and described in good detail. The authors have done well to try to contain the research and limit the wide range of variables that exist in this area of the literature. I think that there are opportunities to identify relevant implications for this work, as these sorts of meta analyses are likely to be useful in driving health advice given by GP’s, and to drive policy change around the provision of outdoor exercise environments. My fear is that researchers will not continue to conduct research in this area given that this review and others have generally found inconclusive or lukewarm support for green exercise.

Response: Thanks for your overall positive comment. We understand your concerns. Notice however that, based on the outcomes of our review, we determined that due to low quality of evidence, future trials are required to evaluate the effects of green exercise. We therefore set a challenge to other researchers in this area to conduct more robust and rigorously designed trials to evaluate the effects of exposure to nature during exercise compared with exercise indoors. We have now emphasized this by adding a section ‘Recommendations for future research’ (p.22, section 4.4)

Point 2: Overall, the article was exceptionally well written, with just a very minor suggestion. On Page 2 line 77- I would normally use ‘associated with’ or ‘linked to’ the notion of…

Response: Line 77 has been edited to “associated with” the notion of…..

Thank you for improving our paper through the constructive comments you have provided. We appreciate the interest you have taken in our manuscript.

Reviewer 2 Report

Thank you for this very comprehensive review about the added benefits of green exercise over non-green exercise. This review is interesting and necessary for program and urban designers making decisions about when, where, and how to improve both programs and neighborhoods for greater population wellness. I would suggest a very few minor revisions.

Your supplemental table 1 has a confusing inconsistency: For the Fuegen study, your control sample seems to be the virtual green sample, while in other studies (e.g., Gatersleben) no non-green alternative is allowed and in yet others (e.g., Plante 2006) Virtual green is allowed as a green exercise condition. I am not sure if that was a mistake or I am interpreting something wrong.

In 3.9 you state that only 4 studies are sufficiently powered for some outcomes. Are all the included studies sufficiently powered for other outcomes? And then in 3.9.1 you report on the findings of many more than 4 studies for those mentioned outcomes, without a clear indication of which are adequately powered. It would be possible to figure it out, I guess, but that is more trouble than most people will go to, so I am concerned that you might be slipping false negatives through the window here. 

Similarly, your harvest plots have a ton of information, and they are very clear - thanks, I suspect that was tough - but it would be helpful to have an indication of adequacy of power on these figures, as well. I know that you go on to discuss some of the challenges of the harvest plot, but that one might be easy enough to rectify with a single symbol or something. Maybe it isn't really an issue, but since you brought up power, you need to address it more comprehensively. Especially since the need for adequately powered studies is one of your final take-away messages.

If you are worried about word count, then worry no longer, because you can drop ALL of section 4.5. It adds absolutely nothing beyond what was already presented in the introduction.

I would appreciate seeing a single-sentence comprehensive synthesis of all the findings. Given that some of these things are better with green exercise, lots are no different, and very few are worse, if green exercise even something worth pursuing? If not, then we don't really need further research on it, right? So I suspect you are still on board ...

Very rarely there are grammatical mistakes. A very thorough editing is necessarily.

Thanks again, and I look forward to sharing this with a bunch of people when it is published.

Author Response

Response to Reviewer 2 Comments

Thank you for this very comprehensive review about the added benefits of green exercise over non-green exercise. This review is interesting and necessary for program and urban designers making decisions about when, where, and how to improve both programs and neighborhoods for greater population wellness. I would suggest a very few minor revisions.

Your supplemental table 1 has a confusing inconsistency: For the Fuegen study, your control sample seems to be the virtual green sample, while in other studies (e.g., Gatersleben) no non-green alternative is allowed and in yet others (e.g., Plante 2006) Virtual green is allowed as a green exercise condition. I am not sure if that was a mistake or I am interpreting something wrong.

Response: Many thanks for spotting this inconsistency in table 1. We have now amended this so that all green exercise conditions (i.e. outdoor or virtual) are reported in one column and all “non-green exercise conditions” are reported in the next column.

In 3.9 you state that only 4 studies are sufficiently powered for some outcomes. Are all the included studies sufficiently powered for other outcomes? And then in 3.9.1 you report on the findings of many more than 4 studies for those mentioned outcomes, without a clear indication of which are adequately powered. It would be possible to figure it out, I guess, but that is more trouble than most people will go to, so I am concerned that you might be slipping false negatives through the window here. 

Response: Both false negatives and positives are a concern when studies are underpowered, and we highlight this issue in our discussion. Most of the analyses performed in green exercise studies are in effect exploratory analysis, which represents a major challenge to their interpretation. We agree we could be more explicit about which studies and outcomes were powered for. To remedy this, we have added the following information into the review:

Line 344: “None of the longitudinal studies performed a prospective power analysis, therefore, it is unclear whether any were sufficiently powered to detect changes in the outcomes they assessed.”

Line 518: “None of these studies performed a prospective power analysis.”

Line 577: “The five trials included here all lacked a prospective power analysis.”

We also discuss the threat of false negatives and positives in section 4.3:

Line 710: “Most trials had insufficient statistical power (only 4 trials performed a priori power calculation), and a combination of low power, small samples, and lack of controlling for multiple comparisons (only three trials corrected for this) means the prevalence of false positives and false negatives is likely to be high.”

Similarly, your harvest plots have a ton of information, and they are very clear - thanks, I suspect that was tough - but it would be helpful to have an indication of adequacy of power on these figures, as well. I know that you go on to discuss some of the challenges of the harvest plot, but that one might be easy enough to rectify with a single symbol or something. Maybe it isn't really an issue, but since you brought up power, you need to address it more comprehensively. Especially since the need for adequately powered studies is one of your final take-away messages.

Response: Many thanks for your comments regarding the harvest plots. It was indeed difficult to present so much information in one type of plot. Regarding statistical power, as we state in our work “only four of these trials were sufficiently powered to detect at least moderate effects on affective states [41,43], affective valence [52], and power output [49]”. Instead of adding more information to the harvest plots, I have indicated in the texts when studies were powered to detect changes in a particular outcome, I hope this is acceptable. See:

Line 387: “Both Niedermeier et al. [41] and Focht et al. [43] were sufficiently powered to detect a medium effect of green exercise on affective states, whereas Turner et al. [52] was specifically powered to detect at least a 1-unit change in affective valence.”

Line 408: “When interpreting the results below it is important to note that only two trials were powered to detect at least medium changes in affective state outcomes [41, 43].”

Line 492: “One trial [49], powered to detect power output differences between outdoor and indoor cycling, found statistically higher power output and lower time to completion (p’s<.001) in the outdoor trial compared with indoors.”

If you are worried about word count, then worry no longer, because you can drop ALL of section 4.5. It adds absolutely nothing beyond what was already presented in the introduction.

Response: Thank you for the suggestion. But the journal does not have a word limit, and even though earlier reviews were presented in the introduction, we think it is important to emphasize in the discussions how the findings of our review agree and differ to previous reviews.

I would appreciate seeing a single-sentence comprehensive synthesis of all the findings. Given that some of these things are better with green exercise, lots are no different, and very few are worse, if green exercise even something worth pursuing? If not, then we don't really need further research on it, right? So I suspect you are still on board ...

Response: Based on the low quality of evidence, we believe it is not possible to say whether there is a clear benefit of green exercise. Future robustly designed studies may well reveal meaningful benefits of green exercise. We hope researchers will be guided by our findings and recommendations to improve the state of the evidence in this area also. In this light, we have emphasized such recommendations by adding a section ‘Recommendations for future research’ (p.22, section 4.4) As researchers we also aim to conduct studies in this area to severely test the green exercise concept. 

Very rarely there are grammatical mistakes. A very thorough editing is necessarily.

Response: Many thanks we have again proof read the work and edited accordingly.

Thanks again, and I look forward to sharing this with a bunch of people when it is published.

Response: Thank you for your kind words and your constructive review.

Thank you for improving our paper through the constructive comments you have provided. We appreciate the interest you have taken in our manuscript.